# Function and Mechanism of Abscisic Acid on Microglia-Induced Neuroinflammation in Parkinson’s Disease

**DOI:** 10.3390/ijms25094920

**Published:** 2024-04-30

**Authors:** Tingting Han, Yuxiang Xu, Haixuan Liu, Lin Sun, Xiangshu Cheng, Ying Shen, Jianshe Wei

**Affiliations:** 1Institute for Brain Sciences Research, School of Life Sciences, Henan University, Kaifeng 475004, China; 104754211509@henu.edu.cn (T.H.); xuyx@henu.edu.cn (Y.X.); haixuanliu@henu.edu.cn (H.L.); chengxs@henu.edu.cn (X.C.); 2College of Chemistry and Molecular Sciences, Henan University, Kaifeng 475004, China; 3Department of Physiology, Zhejiang University School of Medicine, Hangzhou 310058, China; yshen@zju.edu.cn

**Keywords:** microglia polarization, signal pathway, neuroinflammation, neuroprotection, abscisic acid, Parkinson’s disease

## Abstract

Parkinson’s disease (PD), as a neurologically implemented disease with complex etiological factors, has a complex and variable pathogenesis. Accompanying further research, neuroinflammation has been found to be one of the possible factors in its pathogenesis. Microglia, as intrinsic immune cells in the brain, play an important role in maintaining microenvironmental homeostasis in the brain. However, over-activation of neurotoxic microglia in PD promotes neuroinflammation, which further increases dopaminergic (DA) neuronal damage and exacerbates the disease process. Therefore, targeting and regulating the functional state of microglia is expected to be a potential avenue for PD treatment. In addition, plant extracts have shown great potential in the treatment of neurodegenerative disorders due to their abundant resources, mild effects, and the presence of multiple active ingredients. However, it is worth noting that some natural products have certain toxic side effects, so it is necessary to pay attention to distinguish medicinal ingredients and usage and dosage when using to avoid aggravating the progression of diseases. In this review, the roles of microglia with different functional states in PD and the related pathways inducing microglia to transform into neuroprotective states are described. At the same time, it is discussed that abscisic acid (ABA) may regulate the polarization of microglia by targeting them, promote their transformation into neuroprotective state, reduce the neuroinflammatory response in PD, and provide a new idea for the treatment of PD and the selection of drugs.

## 1. Introduction

PD is one of the most common neurodegenerative diseases, second only to Alzheimer’s disease (AD) in incidence, which is mainly a cognitive and motor disorder caused by structural changes or loss of function of the body’s neurons [1]. The main pathological features of PD are the progressive degeneration of DA neurons in the substantia nigra pars compacta (SNc) of the midbrain, the accumulation of aberrant proteins such as the a-synuclein (α-syn) in the Lewy bodies, and inflammatory responses caused by overreactive glial cell proliferation and accumulation [2]. According to epidemiological research, 7–10 million people worldwide suffer from PD, with a prevalence rate of about 1% at age 60 and an increase to 4% beyond that age [3], and with the aging of the population, the prevalence rate also shows age-dependent characteristics [4]. PD not only puts the health of the senior population at risk, but it also costs their family and society a great deal of money [5], so how to effectively treat PD has become an urgent problem. 

The occurrence of PD is closely related to a variety of factors, leading to the current lack of clarity about the exact pathogenesis of the disease. Multidimensional studies have shown that the occurrence of PD may be related to oxidative stress [6], mitochondrial dysfunction [7], trophic factor deficiencies [8], inflammatory responses [9], genetic factors [10], and other factors. Among them, along with the rapid development of neuroimmunology in recent years, studies have shown that inflammatory effects are closely involved in the occurrence of PD [11], and exploring the inflammatory response in the process of PD has an irreplaceable role in refining the mechanism of PD occurrence. As the innate immune cells in the brain, microglia have the ability to clear and phagocytic substances in the nervous system that potentially threaten brain health [12]; it maintains the healthy state of nerve tissue by phagocytosis and degradation of cell debris, abnormal proteins, extracellular deposits, etc., which has received extensive attention from researchers [12,13,14]. Studies have found that there is an over-activation of microglia in the brain of PD patients, and the activated microglia are polarized into different functional states, and the microglia in the damaged areas of the brain of PD show a neurotoxic state in response to inflammation and release a large number of inflammatory cytokines [15]. It is worth noting that the inflammatory response is two-sided in maintaining the health of the organism. Low levels of inflammation play a positive role in maintaining the body’s health by activating immune defenses to remove harmful substances. However, excessive inflammatory responses can cause damage to normal tissue cells and lead to disturbed microenvironmental homeostasis [16]. 

In PD, there is an imbalance in the ratio of neurotoxic microglia to neuroprotective microglia, which results in high levels of inflammatory factor expression and reactive oxygen species (ROS) release, causing an inflammatory response in the brain, which leads to the collapse of this beneficial defense mechanism and irreversible damage to dopamine neurons, leading to DA neuron death and accelerating the disease process [17,18,19,20]. It is worth noting that the current reports show that the activation and polarization of microglia are regulated by a variety of factors, such as inflammatory cytokines, chemokines, and abnormal aggregation proteins [21,22,23]. By targeting and regulating the polarization of microglia, increasing and promoting the number and function of microglia in neuroprotective state, it is possible to save the neuronal death caused by an excessive inflammatory response, thus alleviating PD symptoms [24]. Therefore, targeting the regulation of the polarization of microglia and their functional state transition is an effective and promising therapeutic target.

Currently, a number of drugs have been developed to inhibit neurotoxic microglia, such as cyclooxygenase (Cox)-1 inhibitors and Cox-2 inhibitors, including indomethacin, NS-398, and ibuprofen which regulate the ratio of two different states of microglia by decreasing the number of neurotoxic microglia [25,26] and reduce the inflammation in PD, thereby alleviating the damaging effects of neurotoxicity on DA neurons, and thus ameliorating the pathology of PD. In addition, in study of anti-neurodegenerative diseases, simple anti-inflammatory strategies have been found to have no significant effect on the symptoms and progression of neurodegenerative diseases [27]. Abscisic acid (ABA) is a phytohormone that regulates important physiological functions in higher plants and plays a variety of roles in plant growth and development, including regulating seed dormancy and germination, and controlling stomatal closure [28,29]. In addition, with further research, the beneficial role of ABA in a variety of diseases is emerging [30,31,32], and thus has received tremendous attention. In this article, we will summarizing the regulatory role of microglia in PD and the mechanism of their transition from a neurotoxic to a neuroprotective state. At the same time, ABA which may have potential therapeutic effects on PD, is summarized, in such a way as to provide assistance in improving the inflammatory microenvironment in the brain, reducing neuroinflammation in the brain of PD patients, slowing down the progression of the disease in PD, and improving both motor and non-motor symptoms in PD patients.

## 2. Microglia Origin and Phenotypes

Microglia are widely distributed in the central nervous system (CNS), and Del Rıó Hortega first proposed that microglia originate from mesodermal cells and that these mesodermal cells invade the brain at a later stage to form microglia as the embryo develops [33]. Although, the origin of microglia remains controversial [34,35]. However, some researchers have confirmed Del Rıó Hortega’s hypothesis through optical microscope analysis and immunohistochemistry [33]. In mice, microglia originate from red myeloid progenitor cells of the yolk sac infiltrated during development [36], migrate to the neuroepithelium via the blood circulation, enter the brain parenchyma, and regulate blood–brain barrier (BBB) permeability and leukocyte extravasation and vascularization, which maintain the stability of the brain microenvironment [37]. Microglia, as resident macrophages in the central nervous system, are an important component of neuroglia, accounting for about 10–15% or so of neuroglia, and have been implicated in the pathogenesis of many neurodegenerative and other inflammatory diseases in the brain [14]. 

Studies have shown that as the inherent immune cells, microglia are mainly distributed in the spinal cord and the brain, and constitute the innate immune system in the body, by changing its own functional state and migrating to the site of infection to participate in various immune responses [38], including the detection of pathogens, the clearance of damaged or apoptotic cells and metabolites and tissue debris [39], and immune-inflammatory responses to defend against invasion by pathogens [12]. Under normal physiological conditions, microglia are highly branched [40], maintain the dynamic balance of neuronal cell pool and participate in neurogenesis by regulating neuronal circuits [41]. In addition, they release cytokines such as tumor necrosis factor-alpha (TNF-α) and interferon-gamma (IFN-γ), which are involved in regulating the communication between neurons and other neuroglial cells and thus mobilize other cells in the brain (e.g., astrocytes [42]) to support neuronal survival and development, scavenge harmful materials, take part in neuronal remodeling, repair, and synaptic pruning [43], and preserve the homeostasis of the brain’s microenvironment [44,45,46]. More importantly, glial cell neurotrophic factor (GDNF) signaling, which is released by microglia, GDNF linked to higher cognitive functions like learning and memory [47], and by phagocytosing apoptotic newborn neurons in the granule cell layer of the dentate gyrus of the hippocampus through the terminal end of the protrusions contributes to the regulatory process of the entire nervous system and promotes synapse formation [48]. However, when inflammation, infection, trauma, or other neurological diseases occur in the brain, they cause disruption of the brain microenvironmental homeostasis [49]. Microglia respond positively to these changes and make a responsive response, at which time microglia morphologically show enlarged cytosol, shorter protrusions, and rounded or rod-shaped cell morphology [50]. Notably, studies of PD microglia have revealed that microglia in the PD brain are polarized to a neurotoxic state, which accelerate the disruption of the blood–brain barrier by increasing the release of cellular inflammatory factors, enhancing phagocytosis, and metabolic disorders that exacerbate the inflammatory environment [51], causing DA neuron damage and exacerbating the course of the disease. Overall, microglia exhibit dual roles in the brain, and exploring how to maintain/regulate the normal physiological function of microglia has great potential in the prevention/treatment of neurological diseases.

## 3. Functions and Roles of Microglia in Different States

In PD, microglia exhibit two functional states, both neurotoxic and neuroprotective microglia [15,52]. Microglia polarization occurs from a relatively resting state to a neurotoxic state in response to lipopolysaccharide (LPS), IFN-γ, and cellular/bacterial debris, thereby exhibiting a high degree of phagocytosis, which allows for the rapid clearance of invading pathogens, as well as the expression of inflammation-associated genes [13], which mediate tissue damage [53]. In addition, the presence of endogenous stimuli in the environment, including aggregated α-syn, mutant superoxide dismutase (SOD), and deposited Aβ and tau oligomers, may also result in neurotoxic microglia that consistently induce an inflammatory response and cause irreversible neuronal loss [54]. Microglia in a neurotoxic state cause BBB injury by releasing pro-inflammatory cytokines and chemokines [55], such as TNF-α, inducible nitric oxide synthase (iNOS), and Chemokine C-C Motif Ligand (CCL)-5, which in turn activate astrocytes. Following BBB injury, chemokines released by microglia induce peripheral circulating immune cells to infiltrate into the brain and further amplify BBB injury caused by inflammation through intercellular interactions that exacerbate the infiltration of lymphocytes into the neuronal injury site, causing an inflammatory response in local tissues, leading to tissue damage and apoptosis [56]. In addition, neurotoxic microglia induce the death of DA neurons in the SNc [57], which intensifies the substantial damage to nearby neurons, worsens motor impairments in PD patients, and encourages neurodegenerative disease [58], creating a vicious loop between dying neurons and neuroinflammation.

Neuroprotective microglia are a group of immune cells with the ability to modulate the inflammatory state of the immune system, and they exhibit a significant protective role in neurodegenerative diseases caused by excessive inflammatory responses. Physiologically, neuroprotective microglia can regulate the inflammatory process in the brain by releasing cytokines and chemokines, and they can also maintain the homeostasis of the brain microenvironment by phagocytosis to clear the potential threatening substances in the microenvironment [15]. In addition, studies have shown that neuroprotective microglia play a role in allergic reactions, parasite clearance, inflammation suppression, tissue remodeling, and immunomodulation [59]. Also, neuroprotective microglia promote angiogenesis [60] and support neuronal survival via neurotrophic factors [54]. It is worth noting, however, that although microglia exhibiting neuroprotective functions in PD safeguard neuronal survival and slow down the disease process in neurodegenerative diseases, microglia with this functional state may play a deleterious role in cancer progression. In the most common and aggressive malignant primary brain cancers, microglia and macrophages in the microenvironment of glioblastoma (GBM) account for about 30% of the tumor tissue, and they play an important role in tumor progression by secreting a number of chemokines and their cognate receptors, which promote tumor growth, value addition, invasion, and angiogenesis [61,62,63,64]. Studies have shown that the chemokine ligand–receptor system as an inflammatory mediator has been found to be inherent to the microglia/macrophage inflammatory cell lineage, whereas the CCR5 receptor among the chemokine receptors is expressed in tumor mesenchymal cells in GBM [65]. By blocking CCR5 receptor expression, neuroprotective microglioma genesis can be prevented, which in turn reduces gene expression and function of neuroprotective microglial cell cytokines, and may potentially reduce the predicted tumor growth [63]. In addition, a study by Wenwen Xu et al. demonstrated that neuroprotective microglia promote the onset of non-small cell lung cancer cell migration and invasion, and enhance their proliferation and anti-apoptotic capacity [66]. Therefore, it is necessary to closely integrate the pathological environment characteristics of the disease in the process of targeting and regulating microglial functional status, so as to maintain microglial cell homeostasis in the brain microenvironment.

Overall, a common characteristic of a number of neurodegenerative illnesses is neuroinflammation driven by neurotoxic microglia; on the other hand, neuroprotective microglia mitigate the inflammatory consequences of neurotoxicity by producing anti-inflammatory mediators (Figure 1). In addition, we list other cytokines and chemokines associated with microglia and their effects on neurons through an additional table (Table 1). Microglia are hence a two-edged sword that is crucial in neurodegenerative illnesses. When aiming to treat/prevent neurological diseases by modulating the polarization of microglia and the shift of their functional state, it is necessary to further target the functional state of microglia in conjunction with the characteristics of each disease.

## 4. Pathways Involved in Altering Microglia State on PD

Studies point to the existence of multiple pathways involved in changes in different states of microglia [67,68,69,70,71,72,73,74,75]. However, the modulation of different pathways induces different cellular states. Among them, the inhibition of toll-like receptor (TLR), Notch, rho-related protein kinase (ROCK), and nuclear factor-κB (NF-κB) pathways promote the transition from a neurotoxic to a neuroprotective state in microglia (Figure 2), while the activation of mitogen-activated protein kinase (MAPK), AMP-activated protein kinase (AMPK), and peroxisome proliferator-activated receptor-γ (PPAR-γ) pathways can also achieve the above-mentioned effects (Figure 3). Notably, there exist some pathways, such as phosphatidylinositol 4,5 bisphosphate 3 kinase/protein kinase B (PI3K/AKT) and Janus kinase/signal transducers and activators of transcription (Janus kinase/signal transducers and activators of transcription (JAK/STAT)) pathways, that play a dual role in regulating microglia polarization and its related functional states (Figure 2 and Figure 3).

### 4.1. Inhibition of Pathway Activity Induces a Shift from a Neurotoxic to a Neuroprotective State in Microglia

The notch signaling pathway is a highly conserved intracellular signaling pathway [76], consisting of four receptors and five ligands. The four receptors are Notch1, Notch2, Notch3, and Notch4, consisting of the intracellular domain, transmembrane domain, and extracellular domain; the five ligands are expressed by cluster of differentiation (cluster of differentiation (CD)4+ T cells, and two different families of Notch ligands in mammals, which are type I transmembrane proteins similar to receptors, Delta-like ligands, and Jagged ligands [77,78,79]. The endoplasmic reticulum produces the Notch transmembrane receptor, which is then carried to the plasma membrane where it interacts with transmembrane ligands on neighboring cells to activate the Notch signaling cascade and induce endocytosis, which modifies the receptor’s conformation [78]. Next, receptor subunit segregation and transmembrane subunit protein hydrolysis release the Notch intracellular structural domain [80], which go to the nucleus and function as a transcriptional co-activator to enhance the expression of target genes. This regulates a range of cellular activities, including microglial state, proliferation, differentiation, death, and stem cell maintenance [81]. In response to the activation of the Notch signaling pathway by different factors in post-ischemia reperfusion, microglia polarize into different states and exert dual effects. Microglia were polarized to a neurotoxic state when the expression level of their downstream molecule Hes1 was increased, whereas microglia were polarized to a neuroprotective state when the expression level of the downstream molecule Hes5 is increased [82]. In studies targeting PD, it was found that the inhibition of the Notch pathway during microbial alterations in a Methyl-4-phenyl-1,2,3,6-tetrahydropyridine (MPTP) mouse model reduced the release of pro-inflammatory mediators from microglia that were overly responsive to environmental changes, and promoted the increased expression of microglia-specific molecules in a neuroprotective state, thereby repairing damaged neurons and slowing the progression of PD [67,83]. This suggests that this pathway plays a role in PD in promoting the release of pro-inflammatory cytokines from microglia, leading to increased neuroinflammatory responses, exacerbating DA neuronal death, and causing neurodegeneration. Thus, the Notch pathway may be an important target for regulating microglial cell polarization and inflammatory responses in PD.

TLR receptors (TLRs) are a family of transmembrane pattern recognition receptors that are highly expressed in microglia [84], characterized by extracellular leucine-rich repeat structural domains involved in ligand recognition by PAMP/DAMP and intracellular toll/IL-1 receptor-like (TIR) domains that initiate downstream signaling [85]. More than 20 families of TLR receptors have been identified and are widely distributed in immune cells, including macrophages, dendritic cells, B cells, T cells, fibroblasts, and epithelial cells [86]. In immune cells such as microglia, TLRs are highly expressed, and they constitute the first immune barrier in the body by detecting invading pathogens and initiating innate and adaptive immune responses, which enables the body to defend itself against external infections [87]. In addition, they also activate the interferon regulatory factor 3 (IRF-3) and NF-κB-dependent signaling pathways involved in physiological processes such as cell proliferation, differentiation, and apoptosis [88]. Each TLR can bind to different ligands and mediate immune inflammation by activating the corresponding immune cells. Microglia mainly express TLR2/4 on their surface, which recognizes LPS, heat shock protein, etc. [89]. LPS is a major component of the outer membrane of Gram-negative bacteria [90], and a factor regulating inflammatory mediators [91,92]. In addition, microglia respond to LPS polarized to a neurotoxic state, further causing neuronal damage and exacerbating the PD process. LPS also induces interleukin 1 receptor-associated kinase (IRAK) autophosphorylation by activating the binding of TLR4 to the bridging protein myeloid differentiation factor myeloid differentiation primary response 88 (MyD88) [93]. Phosphorylated IRAK4 and IRAK1 dissociate from MyD88 and interact with tumor necrosis factor receptor-associated factor-6 (TRAF6) and activate the transforming growth factor-β-activated kinase-1 complex 87 (TAK1), thereby activating two major pathways: IKK/IκB/NF-κB and MAPK (including ERK, JNK, p38)/AP-1 signaling pathways [68]. It further promotes the transcription of inflammatory cytokine genes and aggravates neuroinflammation, thereby damaging neurons and speeding up the process of PD [68,94,95].

ROCK is a serine/threonine kinase that is a potential regulator of polarized microglia [96]. Rho is one of the most important members of Rho GTPases, belonging to the Ras superfamily, and is the upstream activator of ROCK, and the Rho/ROCK signaling pathway plays an important role in inflammation [69]. Studies have shown that the activation of the Rho/ROCK signaling pathway plays an important role in inflammation [97,98]. Several recent studies have shown that the activation of the ROCK pathway is associated with MPTP-induced DA neurodegeneration, and that the ROCK inhibitor feudal reduces inflammation by decreasing the expression of the pro-inflammatory factors nitric oxide (NO), interleukin (IL)-1β, IL-6, and TNF-α, and increasing the expression of the anti-inflammatory factor IL-10 in microglia during PD cell model experiments [69,70]. This suggests that the inhibition of ROCK activity modulates microglia polarization and promotes the conversion of microglia from a neurotoxic to a neuroprotective state, thereby reducing DA neuronal damage and slowing down PD progress.

NF-κB is a key transcription factor associated with microglia in a neurotoxic state, and plays a critical role in diseases caused by inflammation [99]. The NF-κB/Rel family includes NF-κB1 (p50/p105), NF-κB2 (p52/p100), p65 (Rel-A), Rel-B, and c-Rel [100], and these factors can interact to form dimers and play an important role in disease progression. The most common of these are heterodimers composed of p50 or p52 subunits and p65, which in turn produce neurotoxic effects and promote pathological processes [101]. Studies have shown that inflammatory stimulation can increase the phosphorylation and proteasome degradation of IκB inhibitory protein, leading to the release of NF-κB and nuclear translocation [102], thus promoting the release of inflammatory factors, increasing the inflammatory response in the brain [103], expanding the damage to DA neurons, and thus aggravating the pathological process of PD. Many modulators have been found to reduce the inflammatory response and slow down the PD process by inhibiting the expression of NF-κB p65/p50, thereby suppressing the transcription of inflammatory genes and driving the polarization of microglial cells, prompting a conversion from a neurotoxic to a neuroprotective state [71].

### 4.2. Increased Pathway Activity Induces a Shift from a Neurotoxic to a Neuroprotective State in Microglia

MAPK is a class of differentially conserved serine and threonine mitogen-activated protein kinases, consisting of p42/44 ERK, MAPK, JNK, and p38 MAPK [104]. p38 MAPK is one of the subclasses of MAPKs and is the most important member of the MAPK family in terms of regulating inflammatory responses. It can regulate the expression of intracellular nuclear gene expression [105] and also transmits signals from the cell surface to the nucleus, thus playing an important role in biological processes such as cellular inflammation [106], oxidative stress [107], and apoptosis [108]. p38 MAPK has two distinct phenotypes, p38 MAPKα and p38 MAPKβ. p38MAPKα is mainly expressed in microglia, while p38MAPKβ is mainly found in astrocytes [58]. The stimulation of inflammatory factors such as IL, LPS, TNF-α, and platelet-activating factor (PAF) can induce p38 MAPK activation, and the mitogen-activated protein kinase signaling pathway activation complex TAK1 can also activate MAPK, which in turn activates and phosphorylates transcription factor AP-1 [71], promoting the activation of neuroprotective microglia and the release of protective cytokines exerts neuroprotective effects and reduces DA neuronal damage [109]. In addition, the activation of MAPK enhances the release of neurotrophic factors such as brain-derived neurotrophic factor (BDNF) [72], which alleviates neural damage in PD.

AMPK is a serine/threonine protein kinase that promotes the expression of nicotinamide phosphoribosyltransferase (Nampt), increases nicotinamide adenine dinucleotide (NAD+) content, and enhances the deacetylase activity of sirtuin 1 (Sirt1), thereby decreasing the acetylation level of NF-κB and inhibiting the expression of several inflammation-related genes [110]. During inflammation, intracellular calcium Ca^2+^ inward flow is increased, which promotes Ca^2+^ binding to the calmodulin CaM to further activate the highly conserved Ca^2+^/CaM kinase cascade [111], including calmodulin-dependent protein kinase-β (CaMKKβ), and activated CaMKKβ further stimulates AMPK phosphorylation. The toxicity of α-syn in nigrostriatal dopamine neurons can be attenuated by modulating AMPK activity [112], which may have implications for the development of neuroprotective treatments for PD. AMPK activates peroxisome proliferator-activated receptor γ coactivator 1α (PGC-1α) and Sirt1, leading to the deacetylation and activation of PGC-1α, and by improving autophagy, reducing endoplasmic reticulum stress and inhibiting the accumulation and activity of NLRP3 inflammasome, regulating the polarization of microglia, promoting their conversion from a neurotoxic to a neuroprotective state, inhibiting the release of pro-inflammatory cytokines, and reducing the inflammatory response [73,74,75]. In addition, activated PGC-1α leads to increased PPAR-γ transcriptional activity. However, the regulation of microglial imbalance by the activation of AMPK in different states is less studied in the treatment of PD, and needs to be further explored.

As a ligand-induced transcription factor of the nuclear receptor family, PPAR positively responds to important responses in the human body such as regulating mitochondrial function, participating in inflammatory processes, influencing redox homeostasis, promoting wound healing, and metabolizing sugars in the blood [113]. Three PPAR isoforms, namely, PPAR-α, PPAR-β/δ, and PPAR-γ, have been studied; these are involved in the regulation of inflammatory processes as well as in many neurodegenerative diseases [114]. PPARs are activated by lipophilic molecules, form heterodimers with chaperones of the retinoid receptor RXR, and interact with the deoxyribonucleic acid (DNA) of the peroxisome proliferator response element (PPRE) sequence elements, thus actively participates in the transcription of genes that regulate various biological reactions, such as inflammatory processes and neuronal protection, so as to maintain normal physiological activities of the body [115]. It was found that PPAR-γ is highly expressed in microglia, and IL-4 induces a neuroprotective state in microglia through the PPAR-γ signaling pathway [116,117]. It has been shown that the activation of PPAR-γ inhibits IκBα degradation, reduces Rel-A (p65) nuclear translocation, disrupts p65 binding to DNA, and blocks the NF-κB signaling pathway, thereby inhibiting the gene transcription of inflammatory cytokines, attenuating the LPS-induced inflammatory response, and promoting the transformation of the functional state of microglia after polarization into a neuroprotective state [118]. This reduces the risk of neurotoxic microglia related to the inflammatory response arising from an imbalance in neurotoxic microglia/neuroprotective microglia, increasing the protective effect on neurons. In addition, the activation of PPAR-γ also exerts anti-inflammatory and antioxidant effects via the MAPK pathway [119]. Collectively, PPAR-γ agonists may exert neuroprotective effects by modulating the expression of genes in the cell survival pathway and the polarization of microglia, and may be a favorable target for the treatment of neurodegenerative diseases such as PD.

### 4.3. Other Pathways Affecting Microglia Status

AKT, also known as protein kinase B (PKB), is an oncogenic protein. AKT has been shown to be an important downstream molecule of PI3K, and the activation of phosphatidylinositol 3-kinase (PI3K) by G protein-coupled receptor (GPCR) or receptor tyrosine kinase (RTK) initiation leads to the phosphorylation of Thr308 and Ser473 residues of AKT [120], which in turn activates AKT. AKT activation depends mainly on PI3K and phosphatidylinositol-dependent kinase (PDK), as well as growth factors, inflammation, and DNA damage, thereby regulating cell proliferation, growth, cycle, apoptosis, and glycogen metabolism [121]. The PI3K/AKT pathway has been shown to be one of the most important intracellular signaling pathways associated with cell resting, proliferation, cancer, and lifespan, regulating cellular activities such as neuronal cell proliferation, migration, and plasticity [122], and the activation of PI3K/AKT inhibits apoptotic protein expression and promotes cell survival. In addition, the PI3K/AKT signaling pathway alters the state of microglia and promotes their proliferation by activating downstream NF-κB and Glycogen synthase kinase-3 beta (GSK-3β). Microglia responding to low levels of PI3K/AKT signaling polarization leads to a neurotoxic state, whereas microglia responding to high levels of PI3K/AKT signaling polarization leads to a neuroprotective state [123], thus high expression of PI3K/AKT reduces inflammation by affecting the polarization of microglia and promoting their conversion to a neuroprotective state thereby alleviating the progression of PD.

The signal composition of the activator of transcription (JAK/STAT) is relatively simple and consists mainly of tyrosine kinase-associated receptors, the signaling tyrosine kinase JAK, and the effect-producing transcription factor STAT [124], which plays an important role in immune and inflammatory responses [125]. JAK is a non-receptor tyrosine protein kinase, consisting of JAK1, JAK2, JAK3, and TYK2, of which JAK1, JAK2, and TYK2 are widely expressed; STAT consists of STAT1, STAT2, STAT3, STAT4, STAT5a, STAT5b, JAK1, JAK2, and TYK2, which are widely expressed, while JAK3 is mainly expressed in lymphocytes and bone marrow; STAT consists of STAT1, STAT2, STAT3, STAT4, STAT5a, STAT5b, and STAT6, and is widely distributed [126,127,128]. The phosphorylated STAT dissociates from the receptor chain, forms a dimer, transfers to the nucleus, and binds to the promoter [129,130], thereby activating downstream gene transcription and participating in cell proliferation, differentiation, apoptosis, and immune regulation, as well as providing conditions for extracellular factors to regulate gene expression [131]. It has been shown that STAT1 and STAT3 mediate the polarization of microglia into being in a neurotoxic state and increase the expression of pro-inflammatory cytokines and chemokines [132], while STAT6 promotes the neuroprotective state of microglia [133]. In addition, studies have shown that inhibitors of cytokine signaling (SOCS) family proteins can act as negative feedback regulators of the JAK/STAT signaling pathway [125], that SOCS1 and SOCS3 bind to JAK and inhibit STAT1 and STAT3 responses to cytokines, and that the upregulation of SOCS1 and SOCS3 expression can attenuate the inflammatory response by affecting the polarization of microglia and promoting their conversion from a neurotoxic to a neuroprotective state [134,135]. Thus, the dysregulation of JAK/STAT in PD and its role in multiple inflammatory pathways makes it a promising PD therapeutic approach.

Nuclear factor E2-related factor 2 (Nrf2) is a member of the Cap’n’ collar (CNC) family of transcription factors involved in a variety of physiological responses in the body, including redox signaling, xenobiotic metabolism, antioxidant, and anti-inflammatory responses [136]. Multiple amino acids comprise the Nrf2 protein, which exists in seven functional domains, Neh1-Neh7, and maintains normal body activity through interactions. The Neh2 structural domain consists of seven lysine residues responsible for ubiquitin splicing, in addition to two peptide binding motifs (DLG and ETGE) that interact with Kelch-like ECH-associated protein 1 (Keap1) responsible for Nrf2 ubiquitination and its proteasomal degradation under normal physiological conditions [137]. Nrf2 reduces dopamine neuron degeneration in neurodegenerative diseases by upregulating antioxidant genes, inhibiting microglia-mediated neuroinflammation, and ameliorating abnormal mitochondrial function [136]. Therefore, Nrf2 activation may be a new therapeutic approach to target the pathogenesis of PD. However, whether Nrf2 activation-mediated inhibition of inflammation is associated with a shift in the polarized functional state of microglia remains to be further verified.

## 5. Potential Mechanisms of Abscisic Acid on Microglia Functional State for the Treatment of PD

Abscisic acid (ABA) is a 15-carbon sesquiterpene compound with the formula C15H20O4, which contains an asymmetric carbon atom to form an enantiomer, cis and trans [138]. Compared with left-handed compounds, right-handed compounds have stronger biological activity but degrade faster [139]. Its physiological activity depends on the presence of free carboxyl groups, a double bond at alpha- or beta- on the cyclohexane ring, and a double bond at C-2 [140]. The biosynthesis of ABA in plants consists of two pathways: the terpenoid pathway, which directly syntheses ABA from mevalonate (MVA); and the carotenoid pathway, where ABA is indirectly synthesized through the subsequent oxidative cleavage of all-trans C40 carotenoid precursors [141]. ABA is a sesquiterpene signaling molecule found in plants, fruits (figs, blueberries, apricots, bananas, etc.), and mammals [142]. Studies have shown that in plants, ABA regulates plant growth, development, and environmental stress responses [143,144]. And in human cell experiments, ABA has been found to regulate processes such as stem cell expansion [145], release of pro-inflammatory factors [91,146,147], insulin release [148], and glucose uptake [149]. In addition, as an endogenous hormone, ABA is abundantly present in animal brain, plasma, and other tissue sites such as adipocytes, pancreatic β-cells, keratinocytes, granulocytes, monocytes, macrophages, mesenchymal stem cells, and hippocampal neurons [150]. More importantly, ABA can be produced and released by the brain tissue itself, such as microglia in the brain [142,151]. In addition, through further understanding, we found that ABA can act as an anti-inflammatory factor in a variety of diseases by exerting anti-inflammatory effects and alleviating the pathological features and course of the disease. Treatment with ABA in mice models of obesity, diabetes mellitus, colitis, and lung disease resulted in inhibition of neuroinflammation, enhancement of the immune response, promotion of neurogenesis, enhancement of synaptic plasticity, and enhancement of learning, memory, and cognitive abilities of the central nervous system, which led to improvement of disease pathology and alleviation of disease progression [152,153,154]. In addition, ABA plays a vital role in improving depression and anxiety [155].

### 5.1. Beneficial Role of ABA in Inflammatory Diseases

It is worth noting that ABA can influence the course of type II diabetes. It has been shown that ABA can control the metabolic response to glucose and inhibit the development of diabetic pathology by stimulating glucose uptake in skeletal muscle and adipose tissue and increasing energy expenditure in brown and white adipose tissue through a non-insulin-dependent mechanism [30]. In addition, AMPK was further activated by ABA intake and stimulated GLUT4-mediated muscle glucose uptake, with a browning effect on white adipocytes; ABA also improved glucose tolerance in normal and critical subjects, and blood glucose, lipids, and morphometric parameters (waist circumference and body mass index) in critical subjects, and can be used in prediabetes and metabolic syndrome treatment [31]. These results demonstrate the potential therapeutic role of ABA in diabetes. In a rat model of high-fat diet-induced neuroinflammation, chronic treatment with ABA exerts neuroprotective effects and further restores cognitive function by decreasing the number of microglia activation and TNF-α levels [156]. Notably, ABA levels are elevated in human atherosclerotic plates, which greatly reduces the process of inflammatory atherosclerotic lesion development [147], whereas dietary supplementation with ABA improves cardiac systolic vascular pressure, reduces F4/80+ CD11b+ macrophage and CD4+ T-cells deposition in the root of the large aorta, and mitigates pathological development of inflammatory atherosclerotic disease [157]. This evidence suggests the possibility of ABA as a new therapeutic agent to ameliorate the associated diseases induced by inflammatory responses. In addition, ABA increased the expression of aortic endothelial carbon monoxide synthase (eNOS), inhibited the expression of vascular cell adhesion molecule-1 (VSAM-1) and MCP-1, and ABA induced a dose-dependent increase in the intracellular concentrations of cAMP and NO, upregulated the expression of eNOS mRNA, and ameliorated the course of disease development in human aortic endothelial cells [157]. These results demonstrate the anti-atherosclerotic and anti-hypertensive effects of ABA.

Overall, the use of ABA in a variety of different disease models has shown beneficial effects, suggesting that ABA is a potential drug candidate. At the same time, we noted that ABA treatment reduced the level of inflammation in the body, and excessive inflammatory response is also one of the pathogenic mechanisms of neurodegenerative diseases such as PD. Therefore, it is necessary to clarify the action targets of ABA in neurodegenerative diseases and its related mechanism pathways, so as to provide evidence for ABA treatment of neurodegenerative diseases, and then provide potential therapeutic drugs for the treatment of PD and other neurodegenerative diseases. Next, we will discuss the potential effects of ABA in neurodegenerative diseases.

### 5.2. The Role of ABA on Microglia Function in Neurodegenerative Disorders

Patients with neurodegenerative diseases have different degrees of cognitive dysfunction, and the main factors affecting learning and memory ability include chronic neuroinflammation [158,159], neuronal apoptosis [160], and altered synaptic plasticity [161]. Activation of PPAR-γ in neurodegenerative diseases may prevent mitochondrial dysfunction, reduce ROS production, increase PGC-1α activity, upregulate LANCL2, maintain mitochondrial membrane potential, inhibit pro-inflammatory cytokines, protect dopaminergic neurons, reduce macrophage infiltration, and improve abnormal protein aggregation (Aβ, α-syn, etc.) [32,162,163,164,165,166,167,168,169]. The above results suggest that PPAR-γ plays an important role in the occurrence and progression of neurodegenerative diseases by improving memory impairment and emotional state and alleviating disease progression. Therefore, we speculated that ABA may also affect the pathological process of PD by regulating the above responses through the activation of PPAR-γ, but its specific role and mechanism still need further experimental investigation. In addition, we have learned that in chronic kidney disease, when PPAR-γ is activated, it can promote the expression of genes such as fatty acid oxidase (FAO) and fatty acid transporter (FATP), and the expression of these genes can increase the oxidation and transport of fatty acids, thus promoting the removal of abnormal proteins and regulating the occurrence and development of diseases [170,171]. Activation of PPAR-γ can also promote the occurrence of autophagy, thus clearing abnormal aggregators and regulating disease progression [172,173].

### 5.3. The Role of ABA on Microglia Function in Parkinson’s Disease

It is worth mentioning that neuroinflammation, oxidative stress, mitochondrial dysfunction, and aging all affect the integrity of neurons and promote the occurrence and development of PD. Alleviating neuroinflammation, oxidative stress, mitochondrial dysfunction, and delaying aging may provide new ideas for PD treatment. Therefore, there is a growing trend to find natural products with antioxidant, anti-apoptotic, and anti-inflammatory properties for the prevention and treatment of PD. Natural products have received attention for their safety and beneficial biological activity. Abscisic acid, as a natural plant hormone existing in a variety of plant and animal tissues, has significant pharmacological activities, including anti-inflammatory effects, which can alleviate the occurrence and development of various diseases [30,174,175,176,177,178]. In addition, studies have shown that ABA can improve and delay aging by enhancing antioxidant activity, anti-apoptosis, and regulating intestinal flora [179].

In PD, the imbalance of microglia in different functional states promotes the release of a variety of pro-inflammatory factors, amplifies the inflammatory response, increases the damage to DA neurons, and accelerates the pathological process of PD [17,18]. Studies have shown that in PD and other diseases induced by inflammation, activation of PPAR-γ may inhibit the activity of the AMPK/NF-κB pathway, regulate the mutual transformation of microglia with different functional states, and then regulate the release of cytokines, improve the level of inflammation, and alleviate the occurrence and progression of the disease [116,117,118,119]. Therefore, after the action of ABA, it may regulate the proportion of microglia in different functional states in the PD brain by affecting the polarization of microglia, promoting their transformation from a neurotoxic state to a neuroprotective state, reducing the release of pro-inflammatory factors, alleviating the inflammation level in PD, thus increasing the protective effect on DA neurons, and regulating the occurrence and development of PD. In addition, in neurodegenerative diseases, activation of the PPAR-γ pathway promotes neuronal differentiation and axonal polarity, and activation of PPAR-γ with thiazolidinedione (TZDs) reduces oxidative stress levels in neurodegenerative diseases, improves mitochondrial dysfunction, alleviates neuroinflammation, reduces neuronal death, and prevents neurodegeneration, thus improving the course of the disease [180]. ABA has a similar chemical structure to TZDs, so in PD, ABA may also protect neurons by playing an antioxidant stress role and alleviating the occurrence and development of disease. More importantly, cell viability and mitochondrial membrane potential were significantly reduced in 6-OHDA-induced PD cell models. However, after ABA, by exerting its antioxidant and anti-apoptotic effects, oxidative stress levels in PD cell models were reduced, cell viability and mitochondrial membrane potential abnormalities were improved, the protective effect on cells was increased, and PD disease progression was alleviated. However, when PPAR-γ was inhibited, the protective effect of ABA on PD cells disappeared [181]. These results all suggest that ABA may play an important role in the activation of PPAR-γ, and provide reasonable evidence for the potential antioxidant, anti-apoptotic, and anti-inflammatory effects of ABA in PD.

In summary, the use of ABA in PD may protect DA neurons by activating PPAR-γ to play anti-oxidative stress, anti-mitochondrial dysfunction, and anti-apoptosis role, and alleviate the development process of PD. More importantly, the activation of PPAR-γ by ABA may reduce inflammation in PD, improve the pathological features of PD, and delay the pathological process of PD by regulating the abnormal proliferation of microglia, modulating the transformation of microglia polarization and their functional state from neurotoxicity to neuroprotection, and exhibiting neuroprotective effects. The possible targeting pathways of ABA in PD are shown in Figure 4. In conclusion, ABA plays an important role in the alleviation of PD-related pathological indicators and treatment, and may provide a new idea for further exploration of PD treatment. However, the specific mechanism pathway of ABA treatment in PD still needs to be further explored, so as to further improve its exact biological mechanism.

## 6. Conclusions

In this review, we listed the pathway mechanisms that may affect the functional status of microglia, and expounded that ABA may regulate the functional changes of microglia in PD by activating PPAR-γ, thus playing an anti-inflammatory role. In addition, the oxidative stress level and mitochondrial dysfunction of PD patients can be alleviated through the regulation of oxidative stress and apoptosis, and the damage of neurons can be alleviated, thus producing potential benefits in the pathological process of PD and providing potential drugs for the treatment of PD. In general, an ABA-induced microglia state from neurotoxic to neuroprotective plays an important role in regulating the changes in pathological indicators of PD, and has positive prospects for it use in the treatment of PD.

## Figures and Tables

**Figure 1 ijms-25-04920-f001:**
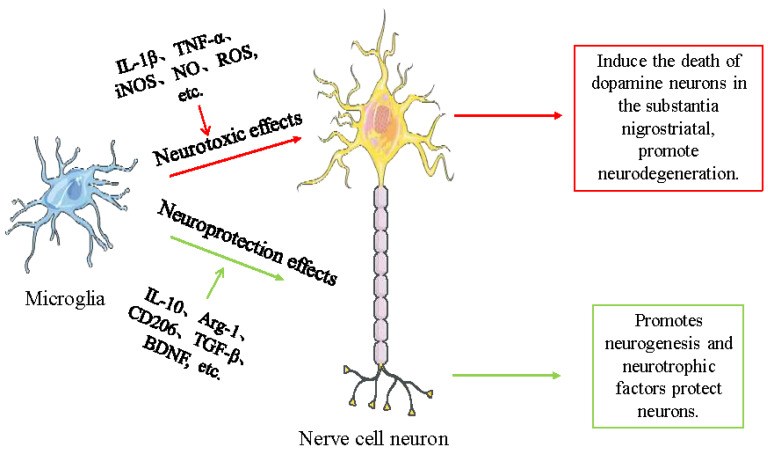
Microglia in different functional states in PD. Microglia responding to different environments exhibit a variety of cellular states, and microglia in PD mainly have neurotoxic and neuroprotective functions. Microglia with neurotoxic function induce dopaminergic neuronal death in the substantia nigra by secreting pro-inflammatory cytokines and chemokines such as IL-1β, TNF-α, and iNOS causing neurodegeneration and aggravating the pathological progression of PD. On the contrary, neuroprotective microglia reduce the inflammatory response, protect neurons, and promote neuronal repair by releasing anti-inflammatory cytokines and neurotrophic factors such as IL-10 and Arg-1. IL: interleukin; TNF-α: Tumor necrosis factor-alpha; iNOS: inducible nitric oxide synthase; NO: nitric oxide; ROS: reactive oxygen species; Arg-1: Arginase-1; CD: cluster of differentiation; TGF-β:transforming growth factor β; BDNF: brain-derived neurotrophic factor.

**Figure 2 ijms-25-04920-f002:**
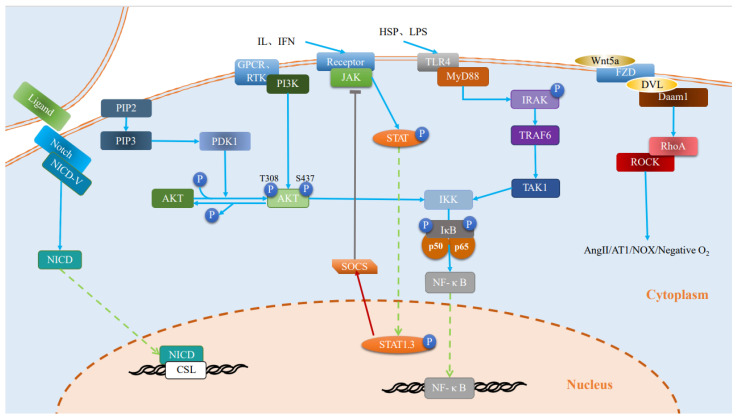
Inhibition of pathway activity induces microglial cell state transition. After activation of Notch by different factors, its released Notch intracellular structural domain enters the nucleus and thus promotes target genes to regulate microglial cell states. Activated TLR4 binds to MyD88 to induce autophosphorylation of IRAK, which interacts with TRAF6. TRAF6 activates TAK1, which activates the pro-inflammatory pathway IKK/IκB/NF-κB to enhance pro-inflammatory gene transcription, and GPCR/RTK initiates activation of PI3K, which interacts with PDK1 to promote phosphorylation of AKT terminal residues, and activated PI3K/AKT further initiates NF-KB, releasing inflammatory factors. The ROCK pathway is activated, releasing toxic factors such as nitrogen oxides, mediating microglia to behave in a neurotoxic state. Activated JAK induces phosphorylation of STAT family proteins, phosphorylated STAT moves to the nucleus and starts the transcription of target genes, STAT1 and STAT3 induce microglia to exhibit a neurotoxic state and promote pro-inflammatory effects and chemokines. IL: interleukin; IFN: interferon; ROS: reactive oxygen species; LPS: lipopolysaccharide; HSP: Heat Shock Protein; NICD:Notch intracellular domain; CSL: CBF-1, Suppressor of hairless, Lag; PIP2: phosphatidylinositol (4,5) bisphosphate; PDK:3-phosphoinositide-dependent protein kinase; PIP2: phosphatidylinositol (4,5) bisphosphate; PIP3: phosphatidylinositol (3,4,5)-trisphosphate; PDK:3-phosphoinositide-dependent protein kinase; PI3K/AKT: phosphatidylinositol 4,5 bisphosphate 3 kinase/protein kinase B; GPCR: G protein-coupled receptor; RTK: receptor tyrosine kinase; JAK/STAT: janus kinase/signal transducers and activators of transcription; SOCS: suppressor of cytokine signalling; TLR: toll-like receptor; MyD88: myeloid differentiation primary response 88; IRAK: interleukin 1 receptor-associated kinase; TAK1:Transforming growth factor-β (TGF-β)-activated kinase 1; TRAF6:tumor necrosis factor receptor-associated factor-6; IKK: ikappaB kinase; IκB: ikappaB; NF-κB: nuclear factor-κB; FZD: Frizzled receptors; DVL: Dishevelled; Daam1:Dishevelled-associated activator of morphogenesis 1; Rho: ras homology; ROCK: rho-related protein kinase.

**Figure 3 ijms-25-04920-f003:**
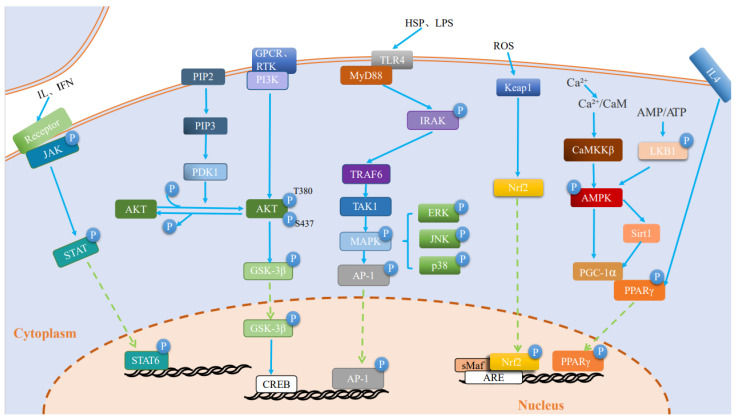
Increased pathway activity induces a shift from a neurotoxic to a neuroprotective state in microglia. Activated complex TAK1 also activates the MAPK pathway, leading to phosphorylation and activation of transcription factor-activated protein-1 (AP-1), and MAPK signaling exerts anti-inflammatory effects and promotes a neuroprotective state in microglia. CaMKKβ). Activated CaMKKβ phosphorylate AMPK plays a role in altering the microglial cell state. Activated JAK induces phosphorylation of STAT family proteins, phosphorylated STAT translocated to the nucleus and starts the transcription of target genes, and STAT6 mediates neuroprotective microglia. Activated PI3K/AKT also activates downstream GSK-3β, which further releases anti-inflammatory cellular mediators. IL-4 promotes microglia in a neuroprotective state via the PPAR-γ signaling pathway. Under oxidative stress (electrophilic or reactive oxygen species conditions), Nrf2 is released from the Keap1-Cul3-RBX1 complex, translocate to the nucleus, and then heterodimerizes with sMaf, leading to its binding to antioxidant response elements (AREs) and transcription of ARE-driven genes. nrf2 reduces DA by upregulating antioxidant genes, inhibiting microglia-mediated neuroinflammation, and improving Nrf2 reduce DA neuronal degeneration by upregulating antioxidant genes, inhibiting microglia-mediated neuroinflammation, and improving abnormal mitochondrial function. IL: interleukin; IFN: interferon; ROS: reactive oxygen species; LPS: lipopolysaccharide; HSP: Heat Shock Protein; CaMKKβ: calmodulin-dependent protein kinase kinase-β; TLR: toll-like receptor; MyD88: myeloid differentiation primary response 88; JAK/STAT: janus kinase/signal transducers and activators of transcription; sMaf: small Maf; PIP2: phosphatidylinositol (4,5) bisphosphate; PIP3: phosphatidylinositol (3,4,5)-trisphosphate; PDK:3-phosphoinositide-dependent protein kinase; ARE: antioxidant response element; PI3K/AKT: phosphatidylinositol 4,5 bisphosphate 3 kinase/protein kinase B; GSK-3β: Glycogen synthase kinase-3 beta; CREB:cAMP response element binding protein; GPCR: G protein-coupled receptor; PGC-1α: peroxisome proliferator-activated receptor-gamma co-activator-1alpha; PPAR-γ: peroxisome proliferator-activated receptor-γ; IRAK: interleukin 1 receptor-associated kinase; TAK1:Transforming growth factor-β (TGF-β)-activated kinase 1; TRAF6:tumor necrosis factor receptor-associated factor-6;MAPK: mitogen-activated protein kinases; AP-1: activating protein-1; RTK: receptor tyrosine kinase; AMPK: AMP-activated protein kinase; ERK: extracellular-signal-regulated kinases; JNK: the c-jun N-terminal kinase; Sirt: sirtuin; Keap1: kelch-like ECH associated protein 1; Nrf2: nuclear factor E2-related factor 2.

**Figure 4 ijms-25-04920-f004:**
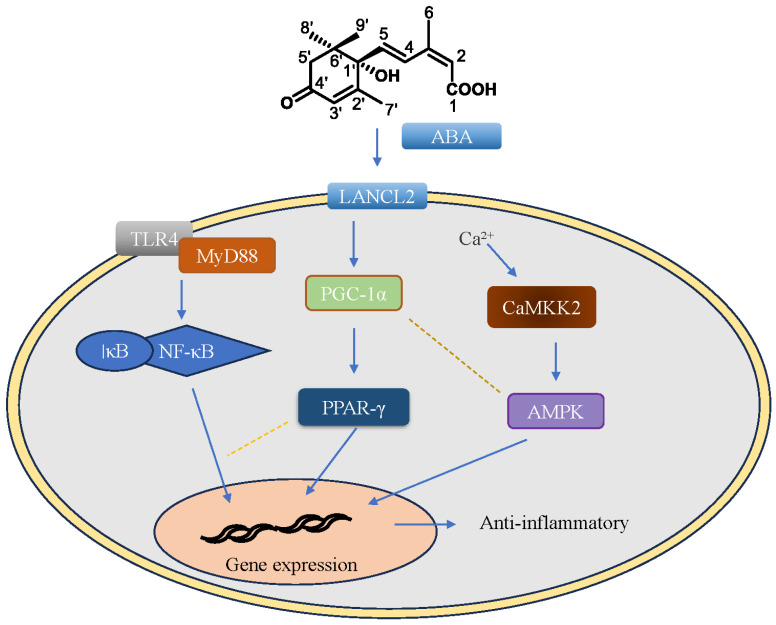
Possible targeting pathway of ABA in PD. After binding with LANCL2, ABA regulates the functional status of microglia and improves the inflammatory response by activating the PPAR-γ pathway. In addition, after ABA, it may also affect the activation of AMPK and NF-κB pathways, co-regulate the state of microglia and the level of inflammation with PPAR-γ, and alleviate the occurrence and development of PD. ABA: abscisic acid; LANCL2: LanC like glutathione S-transferase 2 Gene; TLR: toll-like receptor; MyD88: myeloid differentiation primary response 88; NF-κB: nuclear factor-κB; PGC-1α:peroxisome proliferator-activated receptor-gamma co-activator-1alpha; CaMKK2: calmodulin-dependent protein kinase kinase-2; IκB: ikappaB; AMPK: AMP-activated protein kinase; PPAR-γ: peroxisome proliferator-activated receptor-γ.

**Table 1 ijms-25-04920-t001:** Effects of microglial cytokines and chemokines on neurons.

Cell Phenotype	Cytokines and Chemokines	Effects on Neurons
Neurotoxic Microglia	ROS, RNS (NO)	Increase oxidative stress and mitochondrial dysfunction, promote neuron damage
	CCL1, CCL2, CCL3, CCL5, CCL8, CCL11, CCL12, CCL15, CCL19, CCL20	Recruit T cells, amplify inflammation, promote neuronal dysfunction and death, affect neuronal autophagy
	IL-1α, IL-1β, IL-6, IL-12, IL-15, IL-17, IL-18, IL-23	Increase inflammation, promote neuronal damage, and reduce synaptic plasticity
	TNF-α, M-CSF, IFN-γ	Recruit T cells, further activate microglia, and increase the damage of inflammation to neurons
Neuroprotective Microglia	IL-2, IL-4, IL-10, IL-33	Regulate the abnormal activation of microglia, reduce inflammation, and promote the survival of neurons
	BDNF, GDNF, FGF, NGF, NT-3, NT-4	Induce progenitor cell differentiation, form neurons, promote neuronal differentiation, growth, survival, and axon regeneration
	TGF-β	
	CXCL12, CCL1, CCL14, CCL17, CCL18, CCL22, FIZZ1	Promote cell migration and neuron growth, reduce neuron damage and death
	PDGF, VEGF, IGF1	Promote central nervous system phospholipid repair, inhibit inflammation

## Data Availability

All the data that support the findings of this study are available from the corresponding author upon reasonable request.

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
