# Peer review of "Function and Mechanism of Abscisic Acid on Microglia-Induced Neuroinflammation in Parkinson’s Disease"

_ijms, 2024, doi:10.3390/ijms25094920_

Round 1
Reviewer 1 Report
Comments and Suggestions for Authors
This manuscript, written with a clear logic, summarized the role and research progress of abscisic acid (ABA) in neurodegeneration caused by neuroinflammation, and summarized the mechanism pathways related to ABA regulation of microglia function, thus providing a new strategy for the targeted regulation of neuroinflammation in the treatment of neurodegenerative diseases. In general, this paper is innovative. However, some problems in the article need to be further modified to increase its rigor and readability. After modification, it can be published.
Lines 56-59: Please add references about the effects of different levels of the inflammatory response on human health.
Lines 119-124: Please add references to substantiate your opinion.
Lines 164-166: Replace "Studies" with" Study "or add some references.
Lines 267-272: Please add references to substantiate your opinion.
Lines 297-304: The sentence is too long, please adjust the word order and cite references to explain the point.
Lines 504-556: Please supplement the literature on the beneficial effects of abscisic acid in neurodegenerative diseases, Alzheimer's disease and major depressive disorder…
Lines 534-540: The sentence is too long, please adjust the word order.
Line 550 and 553: Replace "NF-KB" with" NF-κB ".
Comments on the Quality of English LanguageMinor editing of English language required.
Reviewer 2 Report
Comments and Suggestions for Authors
Review report
Function and mechanism of abscisic acid on microglia-induced neuroinflammation in Parkinson's disease
The manuscript under review comprehensively explores the pathways implicated in Parkinson's disease (PD) neuroinflammation, with a particular focus on microglial involvement. It discusses various pathways that influence microglia, potentially leading to neurotoxic or neuroprotective outcomes. Furthermore, it examines the potential therapeutic role of abscisic acid (ABA) in modulating microglial function to counteract neuroinflammation in PD and other neurodegenerative conditions. While the manuscript is generally well-written, there are several areas that could benefit from improvement.
Abstract: The abstract suggests that plant extracts have mild effects and are potentially beneficial in treating neurodegenerative disorders. However, it should be noted that not all natural products exhibit mild effects; some can be toxic, thus limiting their suitability for central nervous system disorders.
Line 51: The phrase "clearing abnormal substances" in line 51 requires clarification.
Line 73: The names of the inhibitors should be mentioned here for completeness.
Line 76-78: The statement regarding drug dosage in lines 76-78 lacks context. Since the drugs mentioned are likely nonclinical, specifying the relevance of dosage in the treatment process is necessary for clarity.
# Throughout the manuscript, microglial functions are discussed in terms of their neurotoxic and neuroprotective states. It would be beneficial to consistently relate these states to microglial polarization, enhancing the coherence of the discussion.
Besides Figure 1, it is suggested to include an additional table listing different protective and toxic cytokines and chemokines associated with microglia, along with their effects on neurons.
In Section 5, it is recommended to present the structure of the ABA molecule and include an additional figure summarizing the pathways targeted by ABA.
Furthermore, is is recommded to expand the section on ABA to establish a clearer link between ABA and PD treatment. This is essential for strengthening the manuscript's overall argument.
In Section 5.2, further clarification is needed regarding how PPAR-γ activation reduces the aggregation of abnormal proteins. Referencing relevant literature on PPAR-γ activation and its impact on protein aggregation would strengthen this aspect.
Conclusion: The conclusion should be revised following the eexpansion of the section on ABA to better justify the focus on ABA for PD treatment via microglial modulation.
Comments on the Quality of English Languagesome editing is required
Round 2
Reviewer 2 Report
Comments and Suggestions for Authors
Adequately addressed most of the comments but there remains one unaddressed comment
# Throughout the manuscript, microglial functions are discussed in terms of their neurotoxic and neuroprotective states. It would be beneficial to consistently relate these states to microglial polarization, enhancing the coherence of the discussion.
Comments on the Quality of English LanguageNone
